# Methane Contributions of Different Components of *Kandelia candel*–Soil System under Nitrogen Supplementation

Chen Feng [1,2], Huiming You [2,3,*], Fanglin Tan [2,3], Jianliang Han [2], Xiaoxue Yu [4], Weibin You [1] and Dongjin He [1,*]

1   Landscape Geography and Ecological Resources Process Research Group, College of Forestry, Fujian Agriculture and Forestry University, Fuzhou 350002, China; fjndcf111@gmail.com (C.F.); wbyou@fafu.edu.cn (W.Y.)
2   Fujian Academy of Forestry, Fuzhou 350012, China; fanglintan@163.com (F.T.); hjlsdau@163.com (J.H.)
3   Wetland Ecosystem Research Station in Quanzhou Estuary, Quanzhou 362000, China
4   Qinling Ecological Protection Center of Hantai District, Hanzhong 723000, China; fjffchen111@gmail.com
*   Correspondence: huimingyou99@gmail.com (H.Y.); fjhdj1009@fafu.edu.cn (D.H.)

**Abstract:** *Kandelia candel* is the most widely distributed tree species on the southeast coast of China and is also the main afforestation tree species along the coastal wetland. In recent years, inorganic nitrogen pollution has become increasingly severe, and investigating the effects of nitrogen input on methane emissions in *Kandelia candel*–soil systems has become significant from a global change perspective. However, the effect of nitrogen input on methane emissions in coastal wetland systems is still uncertain. The field tidal environment is complex and varied, and thus it is difficult to accurately control the amount of nitrogen in the system. Therefore, in order to accurately assess the effects of different concentrations of foreign nitrogen input on methane emission fluxes in a *Kandelia candel*–soil system, we use indoor tidal simulation experimental devices and design two simulation systems with and without plant planting to explore the difference of methane emission flux in this system under five nitrogen input concentrations: N0 (0 g N·m$^{-2}$·a$^{-1}$), N1 (5 g N·m$^{-2}$·a$^{-1}$), N2 (10 g N·m$^{-2}$·a$^{-1}$), N3 (20 g N·m$^{-2}$·a$^{-1}$), and N4 (30 g N·m$^{-2}$·a$^{-1}$). The results showed that: (1) The introduction of *Kandelia candel* promoted methane emissions in coastal wetland ecosystem. Under each nitrogen application concentration, the mean CH$_4$ emission flux in the planting group was 42.98%, 65.59%, 40.87%, 58.93% and 39.23% higher than that in the non-planting group, respectively. (2) Nitrogen input significantly promoted methane emissions in both planted and non-planted environments, and the promoting effect showed as follows: N4 > N3 > N2 > N1 > N0. (3) After the introduction of *Kandelia candel*, the contribution of *Kandelia candel* and soil microorganisms to methane emissions was different under different concentrations of nitrogen addition. The contribution rate of *Kandelia candel* to CH$_4$ emission flux of *Kandelia candel*–soil system ranged from 10.74% to 60.25%, with an average contribution rate of 37.30%. The changed soil microbes contributed 39.75% to 89.26% to the CH$_4$ emission flux in the *Kandelia candel*–soil system, with an average contribution rate of 62.60%. Under N3 nitrogen application concentration, the emission flux of plant was the largest, which was significantly higher than that of the soil microbial pathway; at other concentrations, the methane emissions from the soil microbial pathway were greater than that of the plant pathway, and the contribution rate to the plant–soil system reached 60.25%. The results of this study provide an important basis for improving the estimation accuracy of carbon emissions in coastal waters and formulating policies for the restoration and protection of coastal wetlands.

**Keywords:** exogenous nitrogen; *Kandelia candel*–soil system; methane; mangrove wetlands

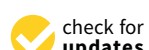



## 1. Introduction

The situation of global change is increasingly serious. CH$_4$ is the second largest greenhouse gas in the atmosphere after CO$_2$, and its contribution rate of greenhouse effect is about 23% [1]. In the past 20 years, the rate of increase in the concentration of CH$_4$ in the

atmosphere is 100 times that of $CO_2$ [2], and the concentration of atmospheric methane is still increasing at a rate of 1.0–1.2% per year [3]. The wetland ecosystem is a huge carbon pool on land and the main natural source of $CH_4$ emissions [4,5]. Recent reports show that half of global methane emissions come from highly variable aquatic ecosystems [6], therefore, tidal flat wetlands play an important role in the study of carbon cycling in global ecosystems [7].

Estuary wetlands are located in the transition zone between land and sea. With the rapid economic development, inorganic nitrogen has become the most important factor exceeding the standard in coastal waters [8]. The ocean tides, atmospheric dry and wet deposition, and inland nitrogen input have made it an enriched area of nitrogen [9], and the input of exogenous nitrogen has been increasing [10–12]. Studies have confirmed that a high nitrogen input will change the system's greenhouse gas emissions and may even exacerbate global warming [13].

As one of the important types of tidal flat wetland, the mangrove wetland is considered to be an effective way to purify pollutants in coastal waters. In the past 20 years, a large amount of sewage and wastewater have been discharged into China's coastal waters without limit, and inorganic nitrogen has exceeded the main standard factor [14]. For mangrove plants in a state of "nitrogen thirst", once an external source of nutrients is input, the growth of mangrove plants will be promoted within a certain range [15]. The effects of the nitrogen input on carbon emissions of a wetland system are complex, including promotion [16,17], inhibition [18,19], or having an insignificant impact [20], which is the result of the combined effects of multiple mechanisms [1,13,16,21–23]. Exactly how nitrogen input affects carbon emissions from mangrove ecosystems is still inconclusive. The field tidal flat environment is complex and changeable, and the input of exotic nutrients is continuously diverse, so it is difficult to carry out experiments to simulate the effect of nitrogen input in the field. As a result, there are few reports on the systematic study of the effect of nitrogen input on the methane emissions in mangrove wetland system at home and abroad. Existing studies have mainly focused on $CH_4$ emissions from existing mangrove wetland ecosystems [24–29], without considering the impact of external sources. Early studies have focused on methane emissions from soils. In recent years, scholars have begun to pay attention to methane emissions in mangrove plants, such as Zhang et al. (2019) and Chen et al. (2018), but these studies are mainly focused on the above-ground parts. Plant–soil microbes are a closely linked with the micro-ecological whole, and methane emissions are closely related to the role of plant root exudates; to estimate the carbon emissions of the ecosystem only based on the carbon emissions from the above-ground parts of plants and the bare land under mangroves may lead to carbon loss or sink. Exactly how nitrogen input affects the carbon emissions of the mangrove-plant–soil system and the contribution of the plant–soil microbe components are still unclear.

In the context of global change, mangrove restoration has been stepped up all over the world, and a large number of mangrove plants have been introduced into tidal flat coastal wetlands. There is no report on the impact of increased external nitrogen input on the system's carbon emissions while promoting the growth of mangrove plants. Ventilation tissue is one of the important ways of methane emission. While nitrogen promotes the increase in plant growth, does its aeration tissue increase correspondingly? Does the flux of methane emissions through plants also increase? Some scholars have found that, while plant biomass increases, the amount of carbon that the soil can store may decrease. This mutual relationship may reduce the soil's carbon sequestration ability [30]. Under the dual-carbon goal, increasing the afforestation of mangrove plants will effectively help to increase blue carbon sinks or increase carbon emissions to a certain extent. This scientific issue needs to be discussed urgently.

Methane emissions in the ecosystem are mediated by microorganisms [4]. The introduction of plants not only changes the composition and activity of microorganisms in the system, but also increases methane emission pathways in the system (plant aerenchyma is also an important pathway of methane emission). Therefore, we rely on the indoor tide simulation laboratory to set up three types of soil system (*K. candel* plant–soil system with plant group, pure soil system with plant group, and pure soil system without plant group), and controlled five nitrogen addition concentrations. Through the simulation study, the following scientific questions are answered: (1) What is the impact of the introduction of mangrove plants on the methane emission of coastal wetlands; (2) What is the regularity of the methane emission intensity of each system in the days after nitrogen addition; (3) How much is the contribution of plants, soil and microbial components to system methane emissions. The results of this study provide a scientific basis for revealing whether the expansion of mangrove afforestation area increases carbon sink or carbon emissions under the background of increasing nitrogen input, and also provides the estimation accuracy of carbon emissions in the near-shore sea and the recovery of coastal wetlands and provides an important basis for the development of protection policies.

## 2. Materials and Methods

### 2.1. Tidal Simulation Experiment

This experiment set three types of systems: *K. candel*–soil system (A1), planting group–soil system (A2, pure soil system with plant group) and no planting group–soil system (CK) (see Figure 1 for details). The methane emission pathways of the ecosystem mainly include soil and plant pathways; therefore, the A1 system in this experimental design reflected the methane emissions of the entire system, including methane emissions from plant pathways and soil pathways. The A2 system, which is the area of pure soil without plants in the planting group, focused on the methane emissions from non-plant pathways after the introduction of plants. The CK system was the methane emission from soil in the absence of plants. The difference of methane emission flux between the A1 system and the A2 system reflected the methane emission from the plant pathway, and the difference between the A2 and the CK systems reflected the methane emission after the change of soil microbial composition and activity. This experiment relied on the tide simulation laboratory of the Fujian Academy of Forestry. The tidal simulation system was composed of an automatic tidal simulation trough device and control system. The automatic tidal simulation trough device is divided into upper trough and lower trough: the upper trough is simulated grass (length × width × height = 1.0 m × 1.0 m × 1.0 m) and the lower trough is storage tank. The medium in the simulation tank was obtained from the wetland of the estuary of Quanzhou Bay, and the water tank contained seawater with a salt concentration of 12% [31]. The simulation tank was connected to the lower tank through a water pump and a drain valve. Since the distribution area of *Kandelia candel* forests in the Quanzhou Bay estuary wetland is mostly half-diurnal tide, the tide form of this experiment as set as half-diurnal tide. A one half-diurnal tide was circulated every 12 h, and two half-diurnal tides were flooded every day for each treatment. The flooding method was submerged flooding, and the flooding duration was 4 h.

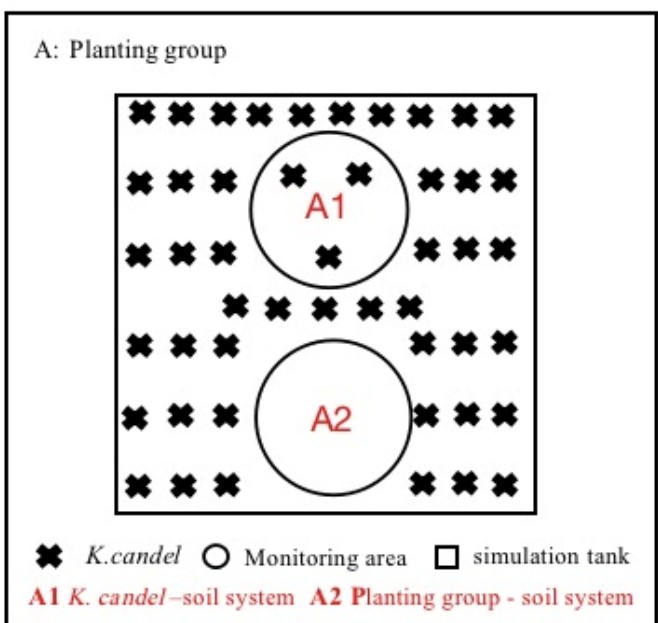
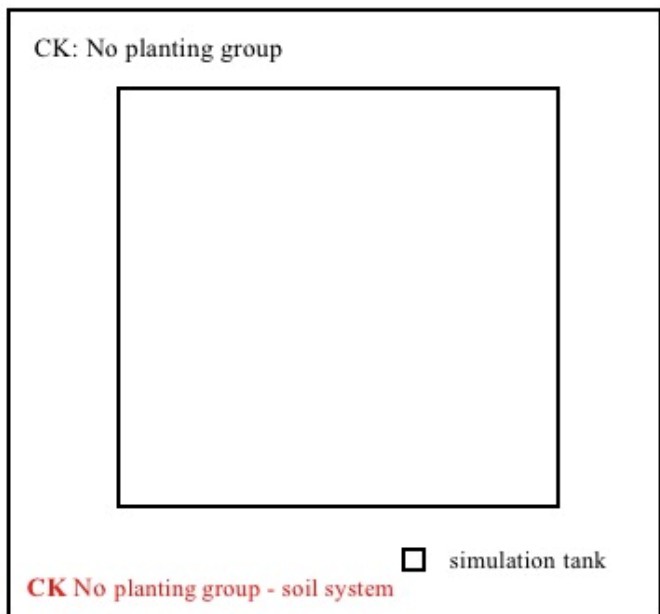

**Figure 1.** Tide simulation system.

### 2.2. Culture Medium, Kandelia candel Nursery and Nitrogen Treatment

The experimental culture medium was taken from the wetland of the estuary of Quanzhou Bay, and a 0–30 cm soil layer was used. Before the start of the cultivation experiment, the system was operated for one month. The soil properties of the undisturbed soil in the simulated experiment are shown in Table 1. The hypocotyl of the simulated seedlings were taken from the estuaries wetland of Quanzhou Bay, and planted in sandy soil in mid-April 2019. At the two-leaf stage, the healthy seedlings with similar size were selected and transplanted into the simulated tank in late May, with 48 seedlings in each simulated trough. The project set up three systems, and five nitrogen input levels, and each treatment had three repetitions. The project set up three systems, five nitrogen input levels, and each group was repeated three times. We used a $NH_4NO_3$ solution to input nitrogen into the simulation tank, and the mean value of total nitrogen deposition in Xiamen, which is close to Quanzhou Bay, is 1.89 g N·m$^{-2}$·a$^{-1}$ (annual nitrogen application per square meter). This research was based on 2 g N·m$^{-2}$·a$^{-1}$ [32], which was multiplied by 2.5 times, 5 times, 10 times, and 20 times, and five nitrogen concentrations were set as follows: N0, N1, N2, N3, and N4, with concentration values of 0, 5, 10, 20, and 30 g N·m$^{-2}$·a$^{-1}$, respectively. From 27 May 2019 to 27 October 2019, nitrogen was applied equally six times, and the same amount of water was applied on 28 June, 30 June, 2 July and 4 July 2019. The corresponding results were measured on 6 July, 8 July, and 10 July 2019.

**Table 1.** Soil physicochemical properties of the simulated experimental soil.

| Volumetric Weight (g/cm³) | TC Content (%) | TN Content (%) | $NH_4^+$–N (mg/kg) | $NO_3$–N (mg/kg) |
|---|---|---|---|---|
| 62.72 ± 0.03 | 1.89 ± 0.13 | 0.13 ± 0.0025 | 3.24 ± 0.34 | 2.71 ± 0.12 |

Note: The data in the table are mean ± standard deviation.

### 2.3. Gas Collection, Measurement, and Calculation

In this experiment, the static box method was used to observe the methane emission flux of the *K. candel*–soil system, planting group–soil system, and the no planting group–soil system during the non-ebb and non-high tide stage. The static box was composed of opaque corrosion-resistant Polyvinyl chloride board with a cylindrical base and a top box. The top box was 35 cm in height, with a built-in electric fan and thermometer. The base was 80 cm in height and 25 cm in diameter. A medical needle was used to collect 100 mL of gas, which

was then pumped into an aluminum foil sampling bag for gas measurements. In order to ensure that the air above the sampling point was fully exchanged with the surrounding air before the next sampling, the gas concentration was restored to the environmental base level. The upper box covering the sampler was opened after each gas collection and closed until the next gas collection. After the collected gas samples were brought back to the laboratory, they were stored in the dark and refrigerated, and the concentration of $CH_4$ in the samples was measured by gas chromatography. The $CH_4$ detector was a hydrogen flame ionization detector (FID), the carrier gas was $N_2$, the flow rate was 30 mL/min, the fuel gas was $H_2$, the flow rate was 30 mL/min, the air was the auxiliary gas, and the flow rate was 300 mL/min. The detector temperature was 250 °C, the separation column temperature was 60 °C, and the inlet heater temperature was 105 °C. The $CH_4$ flux emitted into the atmosphere was calculated according to the following formula [33]:

$$F = \frac{M}{V} \cdot \frac{dc}{dt} \cdot H \cdot \left[ \frac{273}{(273 + T)} \right]$$

Treatments were replicated three times, and the flux data were subjected to analysis of variance and multiple comparison test modules to analyze the differences in $CH_4$ emission fluxes under different environments and different nitrogen treatments using SPSS25.0 (IBM Corp., Armonk, NY, USA).

## 3. Results

### 3.1. The Impact of Kandelia candel on Coastal Wetland CH₄ Emissions

Under different nitrogen concentration treatments, the $CH_4$ emission flux change characteristics of groups with and without plants are shown in Figure 2. In both the presence and absence of plant, the $CH_4$ emission fluxes increased gradually with the increase of nitrogen concentration, and there was a significant difference between the two groups under the five nitrogen concentrations ($p < 0.01$). The mean values of $CH_4$ emission fluxes in both the presence and absence of plant reached a peak at the N4 concentration, and the flux was 0.490 mg·m$^{-2}$·h$^{-1}$ and 0.352 mg·m$^{-2}$·h$^{-1}$, respectively. In the plant group, the maximum amount of $CH_4$ emissions appeared under the N1 concentration (0.099 mg·m$^{-2}$·h$^{-1}$); in the no plant group, the largest increment under N4 concentration treatment was 0.089 mg·m$^{-2}$·h$^{-1}$. Under different nitrogen application concentrations, the mean $CH_4$ emission fluxes in planting group were 42.98%, 65.59%, 40.87%, 58.93% and 39.23% higher than those in the non-planting group, respectively. Therefore, nitrogen input can promote $CH_4$ emissions in wetland ecosystem. Under different nitrogen input concentrations, plant presence promoted $CH_4$ emissions in the wetland ecosystem, and the effect was most significant under N1 nitrogen concentration.

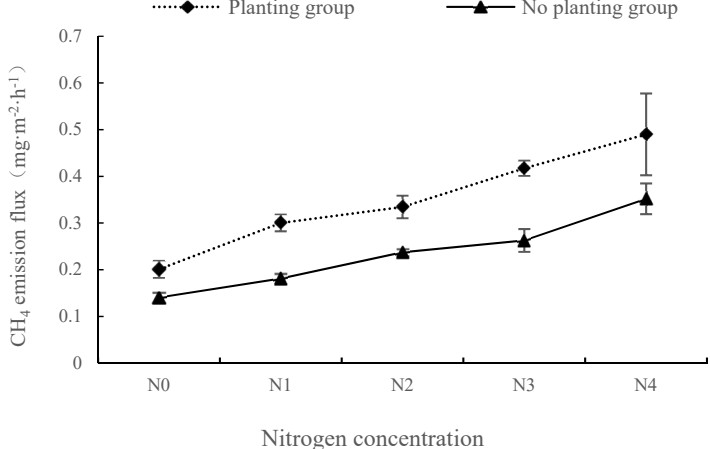

**Figure 2.** Changes of methane emission fluxes under different nitrogen concentrations in plants with and without plants.

### 3.2. CH₄ Emission Rules of Different Systems

The CH$_4$ emission flux of different systems under different nitrogen concentration treatment showed similar volatility change characteristics with culture time (Figure 3). Nitrogen addition promoted the CH$_4$ emission of three systems, and the mean CH$_4$ emission flux at five nitrogen application concentrations was: N4 > N3 > N2 > N1 > N0. Each nitrogen concentration reached the emission peak on the 10th day of culture. Subsequently, the CH$_4$ emission flux of the system at each nitrogen concentration decreased; the CH$_4$ emission flux increases with the culture time, but the promotion begins to decrease after exceeding a period of incubation time. The mean emission of CH$_4$ from the three systems was represented as *Kandelia candel*–soil system > planting group–soil system > no planting group–soil system. In the *Kandelia candel*–soil system, N1, N2, N3 and N4 nitrogen concentrations increased by 60.22%, 75.39%, 136.93%, and 156.17% over the mean control N0 CH$_4$ emission flux, respectively. The one-way analysis of variance revealed that the N0 control and N1 were not significantly different from N 1 and not between N3 and N4 ($p > 0.05$); in others groups, they were extremely significant ($p < 0.01$). For the planting group–soil system, at the N1, N2, N3, and N4 nitrogen input levels, the emission flux of CH$_4$ increased by 38.19%, 57.20%, 77.57%, and 131.08% compared with the control group N0, respectively. The control group N0 showed more significant facilitation than N1, N2, N3, and N4 ($p < 0.01$); no significant difference was observed between N1 and N2 ($p > 0.05$). In the no planting group–soil system, N1, N2, N3, and N4 nitrogen concentrations were increased by 28.99%, 68.93%, 86.90%, and 150.40% over the mean control N0 methane emission flux, respectively. The control group N0 was very significant and different from N2, N3, and N4 ($p < 0.01$), but the control groups N0 and N1 were not significant.

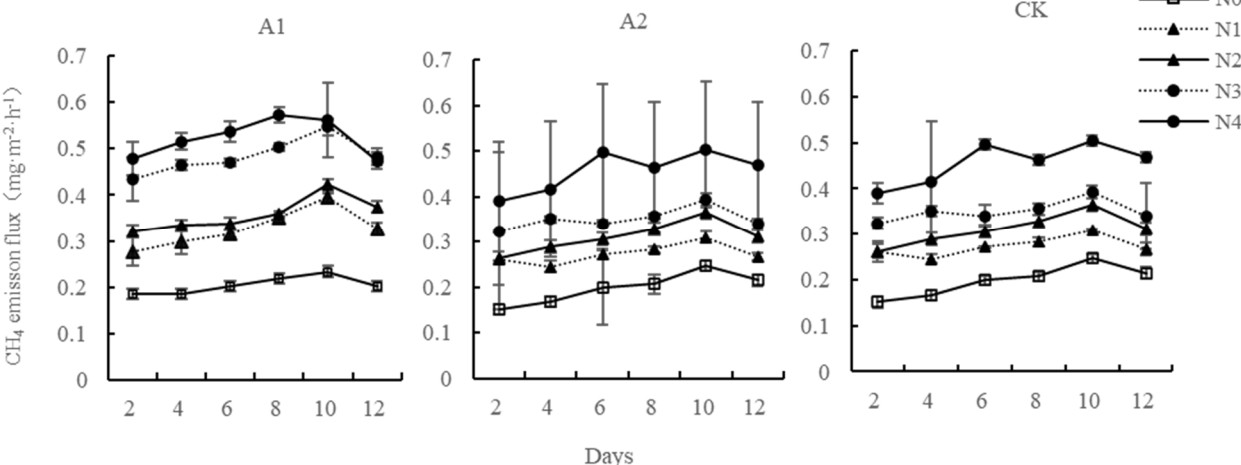

**Figure 3.** Changes of methane emission flux from the *Kandelia candel*–soil system after nitrogen addition. *Kandelia candel*–soil system (A1); Planting group–soil system (A2); No planting group–soil system (CK).

### 3.3. The CH₄ Emission Contribution of Kandelia candel and Microorganisms under Nitrogen Addition

We set three different environmental conditions for the *K. candel* plant–soil system (A1), the planting group–soil system (A2) and the non-planting group–soil system (CK). Through the difference of emissions between the systems, the increased methane emission flux of the plant pathway and the microbial pathway in the soil after the introduction of plants was calculated (See Figure 4). The difference between A1 and A2 reflects the methane produced by the plant pathway, and the difference between A2 and CK reflects the methane emission of soil microorganisms changed by the introduction of plants. The average CH$_4$ emission flux of the A1 is $0.145 \pm 0.081$ mg·m$^{-2}$·h$^{-1}$; the contribution rate of the transmission of the A1 to the CH$_4$ emission flux of the A1 is between 10.74%~60.25%; the average

contribution rate is 37.30%, and the average $CH_4$ emission flux during the cultivation period is $0.062 \pm 0.072$ mg·m$^{-2}$·h$^{-1}$. The contribution rate of methane emission increased by soil pathway due to microbial changes to the $CH_4$ emission flux in the A1 system is from 39.75% to 89.26%, with an average contribution rate of 62.60%, and the average value of $CH_4$ emission flux is $0.083 \pm 0.026$ mg·m$^{-2}$·h$^{-1}$. Under N0 treatment, the increased methane from the soil pathway (0.007 mg·m$^{-2}$·h$^{-1}$) was about 8.314 times that produced by the *Kandelia candel* plant pathway (0.057 mg·m$^{-2}$·h$^{-1}$). There were significant differences between the two approaches ($p < 0.01$). Under N1, N2, and N4 treatments, the methane flux by the *Kandelia candel* plant pathway had a significant increased, but it was still significantly lower than the methane produced by the soil pathway ($p < 0.01$). Different from the N0 control group, the methane gas produced by the three groups of soil microbial pathways was also significantly different from the total methane gas produced by the A1 ($p < 0.01$). We also found that, at the concentration of N3, the methane gas emissions produced by the *Kandelia candel* plant were the largest (0.048 mg·m$^{-2}$·h$^{-1}$), which was significantly higher than the increase in methane emissions from the soil pathway (0.073 mg·m$^{-2}$·h$^{-1}$). This result was different from other nitrogen treatments.

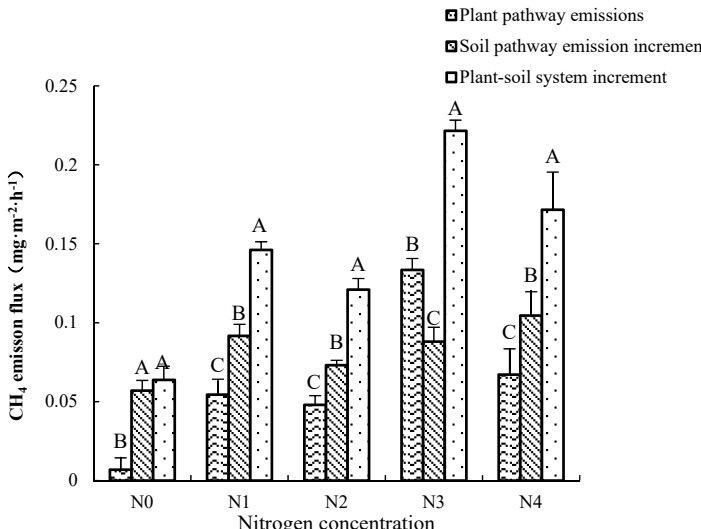

**Figure 4.** Comparison of methane emission fluxes from microbial pathways in plants and soil. Different uppercase letters indicate the difference between different places.

### 3.4. Environmental Factors Analysis

Soil temperature and box temperature and $CH_4$ emission flux were negatively related (Table 2). Among them, there was a significant negative correlation between the box temperature and $CH_4$ emission flux of the planting group ($p < 0.05$), while the negative correlation between the box temperature and $CH_4$ emission of the no planting group was not significant ($p > 0.05$). Except for N0 and N3, the soil temperature in the planting group had a significant negative correlation with $CH_4$ emissions under other nitrogen concentrations ($p < 0.05$). There was no significant negative correlation in the no planting group ($p > 0.05$).

The soil salinity under various nitrogen concentrations showed fluctuating characteristics with the increase in incubation date; there was no significant difference between soil salinity and $CH_4$ emission flux in groups with and without plants ($p > 0.05$). According to the correlation analysis (Table 2), the soil salinity and $CH_4$ emission flux were negatively correlated with the N2 treatment in the planting group, but N2 was positively correlated. In the non-planting group, the soil salinity under N0, N2, and N3 nitrogen concentrations is positively correlated with $CH_4$ emission flux, and negatively correlated with other nitrogen concentrations. Under the environment with and without plants, the correlation between soil conductivity and $CH_4$ emission flux was not significant ($p > 0.05$). Among them, soil

conductivity and $CH_4$ emission flux were positively correlated with N0 and N2 nitrogen concentrations in the planting group. The concentration was negatively correlated, and the performance results were different from those of the planting group. In the non-planting group, there was a negative correlation between soil conductivity and $CH_4$ emission flux at each nitrogen concentration.

**Table 2.** Pearson correlation and significance analysis between environmental factors and $CH_4$ flux.

|  | Nitrogen Concentration | Incubator Temperature (°C) | Soil Temperature (°C) | Soil Salinity (mg/L) | Eh (μs/cm) | $R^2$ |
|---|---|---|---|---|---|---|
| *K. candel* planting group | N0 | −0.874 * | −0.788 | −0.447 | 0.277 | 0.4701 |
|  | N1 | −0.834 * | −0.828 * | −0.334 | −0.101 | 0.5304 |
|  | N2 | −0.829 * | −0.853 * | 0.363 | 0.158 | 0.6449 |
|  | N3 | −0.830 * | −0.675 | −0.304 | −0.040 | 0.5195 |
|  | N4 | −0.835 * | −0.833 * | −0.638 | −0.216 | 0.0372 |
| Non-planting group | N0 | −0.703 | −0.344 | 0.03 | −0.309 | 0.6865 |
|  | N1 | −0.725 | −0.630 | −0.503 | −0.423 | 0.7063 |
|  | N2 | −0.705 | −0.653 | 0.201 | −0.156 | 0.6773 |
|  | N3 | −0.666 | −0.400 | −0.084 | −0.300 | 0.841 |
|  | N4 | −0.703 | −0.675 | 0.319 | −0.089 | 0.720 |

Note: *K. candel* planting group refers to *K. candel*–soil system and planting group–soil system; * indicates a significant correlation ($p < 0.05$).

## 4. Discussion

### 4.1. Kandelia candel Promotes $CH_4$ Emissions from Coastal Wetland Ecosystems

Vegetation plays a role in transporting gas and providing required substrates during the methane emission process of wetlands. Plants provide substrates for methanogens through their root litter and exudates. The $CO_2$ produced by plant respiration and root exudates are an important substrate for methane production [34]. Nitrogen input affects $CH_4$ emissions by promoting plant productivity and biomass. Wetlands transmit $CH_4$ gas to the atmosphere through soil–water–plants, and plants can transmit 50% to 90% of the $CH_4$ produced in the soil to the atmosphere [35]. Some studies have shown that the presence of plants increase the emissions of $CH_4$ flux from coastal wetlands [36,37]. The results are consistent with this study. We found that under five nitrogen concentration treatments, the $CH_4$ emission flux of the plant group was significantly higher than that of the no plant group: as the nitrogen treatment concentration increases, the $CH_4$ emission increases. The cause of this result may have been: (1) nitrogen input promotes the growth of autumn eggplant plants, thus the root exudates in the soil increase, and the organic residues also increase, resulting in an increase in substrates and microorganisms, and an increase in the emission flux of $CH_4$. (2) Nitrogen input promotes plant productivity and biomass in plants, which in turn affects $CH_4$ emissions [38]. The increased biomass can generate the substrates required for $CH_4$ [1], which in turn promotes $CH_4$ emissions. (3) Plants change the soil C/N to change the microbial activity in the soil, accelerate the decomposition of root exudates, and at the same time promote the decomposition of organic matter in the soil, which in turn affects $CH_4$ emissions [37].

In the analysis of the homeostasis data of the soil with and without plants, the research team found that under the condition of nitrogen application, the microbial steady state of the planted group is the sensitive state, while the microbial steady state without the planted group was absolute homeostasis. Additionally, the soil microbial biomass carbon and microbial biomass nitrogen of the planted group were higher than those of the unplanted group. The reason may be that, during the growth of *Kandelia candel* plants, nitrogen input leads to the increase and accumulation of plant root biomass and secretions, which provides more abundant substrates for methanogens; organic substrates are the only source of energy for methanogens with carbon sources, and the activity and function of soil

microorganisms are directly determined by the abundance of substrates [39]. This result is caused by changes in substrates and microbial biomass.

*4.2. Nitrogen Addition Promotes $CH_4$ Emission Fluxes*

This study compared the $CH_4$ emission fluxes of three systems under five different nitrogen concentration treatments, and the results show that the three systems have different degrees of promotion effect on $CH_4$ emissions under different nitrogen input levels, which is expressed as: N4 > N3 > N2 > N1 > N0. There are several possible explanations for this: (1) nitrogen input changes the biomass in the system and affects the supply of C and N available to the soil by plants. Short-term nitrogen input stimulates the activity of soil microorganisms. The organic carbon in the soil can provide substrates for the production of $CH_4$. The abundance of soil directly determines the activity of soil microorganisms and enzymes as well as the performance of their functions, which in turn promoted $CH_4$ emissions [37]. (2) The $NH_4^+$ in the $NH_4NO_3$ solution promoted the production potential of the tidal flat $CH_4$, and $NO_3^-$ also has a promotion of the emissions of the tidal flat $CH_4$. Under the combination of the two solutions, the $CH_4$ emission flux increased after nitrogen input [40]. (3) There are a large number of microorganisms in the soil, and their metabolism and life activities require nitrogen. With the input of nitrogen, the lack of nitrogen content in the original environment was alleviated, and a richer substrate environment was provided for the growth of microorganisms. This improved the activity of the microorganisms in the soil, which in turn promoted $CH_4$ emissions [39,41].

Studies have shown that the average methane emissions from a *K. candel* plant–soil system are 45.41%, 80.62%, 50.97%, 84.34%, 48.76% higher than that of no planting group–soil system at each nitrogen concentration. The cause of this result may have been that nitrogen deposition causes the increased accumulation of root biomass and secretions of *K. candel*, which provides more abundant substrate supply for methane-producing bacteria and increases the amount of $CH_4$ produced in wetlands [42]. *K. candel* have a well-developed aeration tissue and nitrogen input promotes the growth of *K. candel*, increases the number of aeration tissues of the plants, and causes the $CH_4$ emissions of *K. candel* to increase [43]. The mean value of $CH_4$ emissions in the planting group–soil system were 40.54%, 50.56%, 30.78%, 33.52%, and 29.70% higher than that in the non-planting group–soil system under N0~N4, respectively. The reason may be that the presence of the planting group–soil system due to the surrounding *K. candel* plants changed the microbial activity in the soil and stimulated the decomposition of the soil organic matter, while promoting the number of methanogens, resulting in more $CH_4$ [44]. Some studies have shown that nitrogen sinks increase the nitrogen content in wetland soil, which increases the oxidative capacity of wetland soil, thus inhibiting $CH_4$ emissions [10]. The cause of this difference may be due to the difference in the soil, such as differences in microorganisms, organic matter, and pH, resulting in the distinctive difference between the results.

*4.3. Kandelia candel and Soil Microorganisms $CH_4$ Emission Fluxes*

We conducted to distinguish the contribution of plants and soil microorganisms by setting three different environments. There are microorganisms in the three systems; the difference is that the introduction of *K. candel* changed the original microbial environment, resulting in the difference in the final methane emission flux. The contribution rate is 10.74% to 60.25%, and the average contribution rate is 37.40% during the cultivation period. The average $CH_4$ emission flux emitted by the plant pathway, during the period, was $0.062 \pm 0.072$ mg·m$^{-2}$·h$^{-1}$. Chen Kunlong (2018) in the Minjiang River found that the annual contribution of 26.99% to 97.47% was 66.87% [29], while Liu et al. (2011) found only 18.10% in reed wetland [45]. All are within the range of the plant contribution rate of the results of this study, but there are certain differences in the average contribution rate. We can see that different environmental conditions have certain differences between different plant species. The N3 nitrogen concentration had the largest emission flux and contributed 60.25% to the *K. candel* plant–soil system, probably due to the productivity

level and their largest biomass at this concentration, the highest microbial activity in the soil, most conducive to the production, and release of $CH_4$.

The experiment found that there is a certain difference in the emission fluxes of A2 and CK due to the introduction of autumn eggplant plants. The reason may be that the introduction of plants changed the original microbial activity in the soil or increased the microbial biomass. Except for N3 concentration, the microbial pathway was significantly higher than the contribution of soil microorganisms, which was was 39.75%~89.726%, and the contribution rate was 62.60%. The average value of $CH_4$ emission flux in the culture period was $0.083 \pm 0.026$ mg·m$^{-2}$·h$^{-1}$. The emission flux increases with the increase of the nitrogen concentration, indicating that the nitrogen input enhances the microbial activity in the soil, provides it with a more abundant substrate environment, and promotes its growth. Pangala et al. found that the mean $CH_4$ emission flux in rainforest was $0.0329 \pm 0.0078$ mg·m$^{-2}$·h$^{-1}$ and the peatland pellet emission flux was $0.7 \pm 0.5$ g·m$^{-2}$·h$^{-1}$. Both are lower than the $CH_4$ emitted by the soil microorganisms in this study [46], and the results of the study by Tian Dan et al. in the mangrove forests of the Yingluo Harbor in Guangxi indicate that the $CH_4$ emission flux from the soil is 7.30 mg·m$^{-2}$·h$^{-1}$~27.20 mg·m$^{-2}$·h$^{-1}$, which is significantly higher than the results of this experimental study [47]. The substrates that produce $CH_4$ in the soil mainly come from the soil organic substances and organic substances released or imported from the outside. The richness of the substrates will directly affect the emission of $CH_4$ [37]. The difference between the tropical rainforest wetland and the results of this study may be due to the different organic carbon content in the soil. The tropical rainforest has a large annual rainfall and strong soil leaching. A large amount of organic matter in the soil is washed away by rain, while the Yingluogang mangrove wetland in Guangxi is significant. The reason for the higher than the results of this study may be that the temperature is higher than the indoor tide simulation laboratory of this study.

*4.4. The Relationship between $CH_4$ Emission Fluxes and Its Impacting Factors*

Temperature is an important environmental factor affecting $CH_4$ emission in wetland, which mainly affects the methane emissions of ecosystems by affecting plant transport and methane emission capacity and changing soil microbial activity. This experiment shows that the soil temperature, box temperature, and atmospheric temperature are all negatively correlated with $CH_4$ emission flux at each nitrogen concentration. Wang Qing et al. [48] found that the suitable atmospheric temperature for $CH_4$ emissions in the Dongtan wetland in Chongming, China, is between 19 and 31 °C, and the suitable soil temperature for methane emission is between 25 and 27 °C; when the temperature is higher than this in the range, $CH_4$ flux shifts to a lower level. The soil temperature during the cultivation period is between 27.93 °C and 30.75 °C, and the atmospheric temperature is between 28.1 °C and 36.9 °C. It may be that the higher temperature during the cultivation period has suppressed the methane emission of the system to a certain extent.

Salinity is an important environmental factor that affects many ecological processes in wetlands. Soil salinity mainly affects the $CH_4$ emission flux of the system indirectly by influencing plant activities and microbial activities. Salt water will bring a large amount of sulfate. On the one hand, salt water will bring a large amount of sulfate, and when wetland soil is in an anaerobic environment, sulfate will compete with methanogens for substrates, leading to a decrease in $CH_4$ production [49]. On the other hand, a large number of $SO_4^{2-}$ electron receptors accompanying salt water continuously replace methanogens to produce $CH_4$, resulting in a slower $CH_4$ production rate and reduced $CH_4$ emissions [50]. In this study, the $CH_4$ emission flux of the control group in the planting group was significantly negatively correlated with soil salinity and soil electrical conductivity, but there was no significant correlation under other nitrogen concentrations, indicating that the correlation between soil salinity and conductivity and $CH_4$ emission flux was weakened after nitrogen input. Li Dongdong [9] found that the $CH_4$ emission flux was significantly negatively correlated with soil conductivity and soil salinity in both freshwater and brackish wetlands

in the estuarine wetlands of eastern Fujian in southeast China. Poffenbarger et al. [49] also reached a consistent conclusion in estuarine wetlands. We also found that the conductivity of the *K. candel* planting group was lower than that of the non-planting group, which may be caused by the change of microbial activity in the soil caused by plant root exudates or the presence of plants, which can be further studied in the future. Soil conductivity not only affects the production of $CH_4$, but also affects the gas transport of plants. Conductivity is one of the important indicators to measure the condition of soil anaerobic environment. It has been found that, at lower levels of Eh, plant aerenchyma will become developed, thus affecting the transmission of $CH_4$ [51,52].

## 5. Conclusions

*Kandelia candel* plants promote $CH_4$ emissions from coastal wetland ecosystems. With and without plants, the $CH_4$ emission flux gradually increased with the increase in nitrogen concentration, and there was a very significant difference between the two groups under the treatment of five nitrogen concentrations ($p < 0.01$). At each nitrogen concentration, the average $CH_4$ emission flux of the planting group was 42.98%, 65.59%, 40.87%, 58.93%, 39.23% higher than that of the no planting group. Nitrogen input promotes $CH_4$ emissions from wetland ecosystems; under different concentrations of nitrogen input, the presence of plants promotes $CH_4$ emissions from wetland ecosystems, and it is most significant under the treatment of N1 nitrogen concentration.

Different N input concentrations promoted the system $CH_4$ emission: as the nitrogen concentration increases, the promotion effect increases. N1, N2, N3 and N4 treatment increased the average system $CH_4$ emission flux by 44.11%, 67.09%, 102.35%, 145% and 145.54%, respectively. The effect of nitrogen input on the $CH_4$ emission in the *K. candel* plant–soil system was N4 > N3 > N2 > N1, and the emission peak was reached on the 10th day of cultivation at each nitrogen concentration.

The contribution rate of the $CH_4$ emission flux of *K. candel* plant to the *K. candel* plant–soil system is 10.74%~60.25%, with an average contribution rate of 37.40%. The contribution rate of the transport of soil microorganisms to the $CH_4$ emission flux of *K. candel* plant–soil system is between 39.75%~89.26%, with an average contribution rate of 62.60%. Under the N3 concentration, the $CH_4$ emission flux of *K. candel* plants was the largest, which was significantly higher than the methane gas produced by the soil microbial pathway, the methane produced by the *K. candel* plant pathways under other nitrogen concentrations was significantly lower than the methane produced by the soil microorganisms. In the future, it can be further explored whether the introduction of plants changed the activity of the original microorganisms in the soil or the microbial biomass in the soil, or both.

The impact of nitrogen input on $CH_4$ emissions from wetland ecosystems has significant temporal and spatial heterogeneity. After nitrogen input, different observation times and different observation seasons may cause different impact results; there are differences between indoor simulated tide experiments and natural ecosystems, the future research should be expanded and combined with field investigation to conduct comprehensive analysis, and further supplement and refine the research results.

**Author Contributions:** Conceptualization, H.Y. and F.T.; methodology, H.Y. and F.T.; software, C.F.; validation, D.H., W.Y. and H.Y.; formal analysis, C.F. and H.Y.; investigation, H.Y., J.H. and X.Y.; resources, H.Y.; data curation, C.F.; writing—original draft preparation, C.F.; writing—review and editing, H.Y., W.Y. and D.H.; visualization, C.F.; supervision, H.Y.; project administration, H.Y. and F.T.; funding acquisition, H.Y. All authors have read and agreed to the published version of the manuscript.

**Funding:** This work was supported by the Fujian Provincial Public Welfare Scientific Institutions Project (2019R1009-3).

**Institutional Review Board Statement:** No applicable.

**Informed Consent Statement:** No applicable.

**Data Availability Statement:** The data presented in this study are available on request from the coreesponding author.

**Conflicts of Interest:** The authors declare no conflict of interest.

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
