# Peer review of "Methane Contributions of Different Components of Kandelia candel–Soil System under Nitrogen Supplementation"

_forests, doi:10.3390/f13020318_

Round 1

Reviewer 1 Report

The authors have adressed an important topic, the contribution of wetlands to the global methane emissions. To explore this systematically , they explored the impact of different types of nitrogen supplementation on methane emissions in different laboratory experiments. These consisted of three experimental setups, containing among others Kandelai candel, the most widely distributed tree species in the southeast coast of China. 

The experiments were evaluated by different chemical analyses of and basic statistics. The manuscript is relatively well written but a few things would demand further clarification, e.g.: 

Line 75-80: sentence appears incomplete - and is also rather long, try to rephrase and split into two sentences.

Line 111-113: incomplete sentence.

Line 208: 136,136.93% -  ? 

Line 437-438: Specify what kind of additonal analyses are needed. 

General comments: 

1) Define planting group- what kind of plants (or describe at least the ecosystem somewhat more detailed)? 

2) Source, age and health status of plants?

3) Size/volume of the experimental setup.

4) Amount of digits presented in results - is it appropriate to include 4 digits?

5) How high do you think that the contribution of abiotic processes was? Unfortunately you did not perform any kind of microbiological analyses, yet you constantly describe the processes as microbiological - which it is most likely, but do you know for sure that is is 100 %? 

Author Response

Point 1:Define planting group- what kind of plants (or describe at least the ecosystem somewhat more detailed)?

Response 1:planting group , which is the area of pure soil without plants in the planting group), which focuses on the methane emissions from non-plant pathways after the introduction of plants.Regarding the explanation of this system, I have made a more detailed explanation in Experimental Method 2.1, in lines 122-130 in the revised manuscript.

Point 2:Source, age and health status of plants?

Response 2:The hypocotyls of the simulated experimental seedlings were taken from the Quanzhou Bay Estuary Wetland, which were the seedlings of the current year. They were planted in sandy soil in mid-April 2019. After two new leaves were grown, the healthy K.candel seedlings of similar size were selected and grown in 5 transplanted to the simulation tank in late month, In the revised manuscript, lines 154-158, I have made a more detailed description.

Point 3:Size/volume of the experimental setup.

Response 3:The tide simulation system consists of an automatic tide simulation tank device and a control system. The automatic tide simulation tank device is divided into an upper tank and a lower tank.The upper tank is a simulation tank (length*width*height=1.0m*1.0m*1.0m), and the lower The tank is a water storage tank. Lines 136-139 in the revised manuscript, added.

Point 4:Amount of digits presented in results - is it appropriate to include 4 digits?

Response 4:Modified to three decimal places.If it is not correct, I will modify it again.

Point 5:How high do you think that the contribution of abiotic processes was? Unfortunately you did not perform any kind of microbiological analyses, yet you constantly describe the processes as microbiological - which it is most likely, but do you know for sure that is is 100 %?

Response 5:The experiments were conducted to distinguish the contribution of plants and soil by setting 3 different environments. There are microorganisms in the three systems, the difference is that the introduction of autumn eggplant has changed its original microbial environment, resulting in the difference in the final methane emission flux.

In this experiment, three system types were set up: the K. candel–soil system  (A1), the planting group-soil system (A2, pure soil system with a plant group), and the non-planting group-soil system (CK) (see Figure 1 for details). The methane emission pathways of the ecosystem mainly include soil and plant pathways; therefore, in this experimental design, A1 reflects the methane emissions of the entire system, including methane emissions from plant pathways and soil pathways; A2 is the methane emission from pure soil without plants in the planting group. area, focusing on the methane emissions from non-plant pathways after the introduction of plants; CK is the methane emissions from soils without plants. The difference in methane emission flux between A1 and A2 reflects the methane emission of the plant pathway, and the difference between A2 and CK reflects the methane emission after the change of soil microbial composition and activity after the introduction of plants. In the future, we can further explore whether the introduction of plants changed the activity of the original microorganisms in the soil or the microbial biomass in the soil, or both.

Reviewer 2 Report

A title “introduction” should not be put above the first passage? (Subsequently a renumbering of the sections and subsections would be necessary)

The formulas N·m-2 ·a-1 are not quite clear (especially the factor a needs clarification). Explain also the symbols of the formula in the l.172. Not everyone uses the same symbols. So the readers should be helped

How were the sample units selected? Tell about the sampling method or sampling criteria you used

Specify more the “certain limitations” and “certain differences” you mention in l.435 and 436. Make an assessment about the generalizability/reliability of your results and method. Can you suggest a more specific method for future projects (e.g. random sampling or judgment sampling according to certain criteria)?

Author Response

Response to Reviewer 2 Comments

Point 1:A title “introductionshould not be put above the first passage? (Subsequently a renumbering of the sections and subsections would be necessary)

Response 1:The introductory part has been revised to the first section.

Point 2:How were the sample units selected? Tell about the sampling method or sampling criteria you used.

Response 2:We used NH4NO3 solution to input nitrogen into the simulation tank, the mean value of total nitrogen deposition in Xiamen, which is close to Quanzhou Bay, is 1.89gN·m-2·a-1(annual nitrogen application per square metre). This research is based on 2gN·m-2·a-1 , which is multiplied by 2.5 times, 5 times, 10 times, and 20 times, set five nitrogen concentrations as follows: N0, N1, N2, N3, and N4, with concentration values of 0, 5, 10, 20, and 30 gN·m-2·a-1.

Point 3:The formulas N·m-2 ·a-1 are not quite clear (especially the factor a needs clarification). Explain also the symbols of the formula in the l.172. Not everyone uses the same symbols. So the readers should be helped

Response 3:gN·m-2·a-1 means the amount of nitrogen applied per square meter per year.A large number of existing studies have adopted this unit, such as the ninth reference in the revised manuscript.

Point 4:Specify more the “certain limitations” and “certain differences” you mention in l.435 and 436. Make an assessment about the generalizability/reliability of your results and method. Can you suggest a more specific method for future projects (e.g. random sampling or judgment sampling according to certain criteria)?

Response 4:The impact of nitrogen input on CH4 emissions from wetland ecosystems has significant temporal and spatial heterogeneity. After nitrogen input, different observation times and different observation seasons may lead to different impact results; there are differences between indoor simulated tide experiments and natural ecosystems , the future research should be expanded and combined with field investigation to conduct comprehensive analysis, and further supplement and refine the research results. Revised the expression in lines 495-501 of the revised manuscript.

Reviewer 3 Report

Dear Authors,

Your contribution is of high scientific level, good structured, well illustrated, and with highly valuable scientific output.

Some minimal additional explanations and broader interpretation should be useful.

The details are herewith below.

Lines 49-50

Wetland ecosystem is a huge carbon pool on land and 48 the main natural source of CH4 emissions, accounting for about 50-70% of the total natural methane emissions and one-third of the global annual methane emissions,...

Is everything OK in this sentence with calculations?

Line 59-61

As large amounts of sewage and wastewater are discharged unrestrictedly, mangrove wetlands have become the main buffer zone for nitrogen deposits near the coast.

Indeed. Nevertheless, there is no corresponding information in the manuscript. It should be useful to discuss the rate of sewage and wastewater pollution and thus substantiate the experiment.

 Line 133

Kandelia candel nursery and nitrogen treatment

Experiment is interesting. It is understandable that a model experiment with mature shrubs and trees is impossible. However, an organogenesis of the young and mature plants differs, and this can influence a methane release from the wetland. It would be useful to give a corresponding explanation or even present a model.

Lines 294-296

The transmission of CH4 to the atmosphere by aeration tissues is a major method of CH4 transmission by plants, which affects CH4 emissions.

Notwithstanding you gave a link to the References, the statement is doubtable. For what biological purpose can a plant transmit CH4 through its tissue?

The next doubt is that a gas conductivity of plant tissues is many times less compared to the rate of direct CH4 release from the soil.

Conclusions

A role of the sewage and wastewater as a motive of experiment was not reflected.

Author Response

Response to Reviewer 3 Comments

Point 1:Lines 49-50

Wetland ecosystem is a huge carbon pool on land and 48 the main natural source of CH4emissions, accounting for about 50-70% of the total natural methane emissions and one-third of the global annual methane emissions,...

Is everything OK in this sentence with calculations?

Response 1:Lines 49-50 in the original manuscript "...accounts for 50%-70% of natural CH4 emissions" Your question has been deleted after consideration.

Point 2:Line 59-61As large amounts of sewage and wastewater are discharged unrestrictedly, mangrove wetlands have become the main buffer zone for nitrogen deposits near the coast.

Indeed. Nevertheless, there is no corresponding information in the manuscript. It should be useful to discuss the rate of sewage and wastewater pollution and thus substantiate the experiment.

Response 2:As one of the important types of tidal flat wetlands, mangrove wetlands are considered to be an effective way to purify pollutants in coastal waters.A large amount of sewage and waste water are discharged unrestrictedly. In the past 20 years, inorganic nitrogen has been the most important factor exceeding the standard in China's coastal waters.In this study, a related research and analysis was conducted on the impact of excess nitrogen on methane emissions from the coastal wetland ecosystem in southeastern China, where the K.candel is the main tree species. Therefore, according to the content of the bulletin, the original text has been revised and improved. which can be seen in the revised manuscript 64- 68 lines.

Point 3: Line 133

Kandelia candel nursery and nitrogen treatment

Experiment is interesting. It is understandable that a model experiment with mature shrubs and trees is impossible. However, an organogenesis of the young and mature plants differs, and this can influence a methane release from the wetland. It would be useful to give a corresponding explanation or even present a model.

Response 3:We used the K.candel seedlings born in the current year, which had been cultivated for more than a month when transplanted into the simulation tank. The selected K.candel seedlings were all healthy and similar in size. Different growth stages of the plants may indeed have a certain impact on the research results. This will be expanded and extended in future research.

Point 4:Lines 294-296.The transmission of CH4 to the atmosphere by aeration tissues is a major method of CH4 transmission by plants, which affects CH4 emissions.

Notwithstanding you gave a link to the References, the statement is doubtable. For what biological purpose can a plant transmit CH4 through its tissue?

Response 4:Many related studies have shown that most of the methane is absorbed by the root system, and then transported into the atmosphere through the ventilated tissue, which is a form of methane transport by plants. The elaboration of the analysis of the results is not suitable, therefore, I have revised this point in lines 326-329 of the revised manuscript.

Point 5:The next doubt is that a gas conductivity of plant tissues is many times less compared to the rate of direct CH4 release from the soil.

Response 5:The conductivity of the planting group was lower than that of the non-planting group. The result may be caused by the presence of plant root exudates or the presence of plants changing the microbial activity in the soil, which can be further studied in the future. Soil conductivity not only affects the production of CH4, but also affects the gas transport of plants. Conductivity is one of the important indicators to measure the condition of soil anaerobic environment. It has been found that at lower levels of Eh, plant aerenchyma will become developed, which in turn affects the transmission of CH4. For the supplementary content, see lines 402-408 of the revised manuscript, and I have also included relevant references in the revised manuscript.
